# Hygiene Practices Against Dermatophytic Fungi: A Review of Strategies to Combat Antifungal Resistance

**DOI:** 10.3390/biology14081016

**Published:** 2025-08-07

**Authors:** Aditya K. Gupta, Daniel Taylor, Tong Wang, Elizabeth A. Cooper, Ditte Marie L. Saunte

**Affiliations:** 1Division of Dermatology, Department of Medicine, Temerty Faculty of Medicine, University of Toronto, Toronto, ON M5S 1A1, Canada; 2Mediprobe Research Inc., London, ON N5X 2P1, Canada; dtaylor@mediproberesearch.com (D.T.); twang@mediproberesearch.com (T.W.); lcooper@mediproberesearch.com (E.A.C.); 3Department of Dermatology, Zealand University Hospital, 4000 Roskilde, Denmark; ditte.marie.saunte@regionh.dk; 4Department of Clinical Medicine, Faculty of Health and Medical Sciences, University of Copenhagen, 2100 Copenhagen, Denmark; 5Department of Dermatology and Allergy, Copenhagen University Hospital—Herlev and Gentofte, 2900 Gentofte, Denmark

**Keywords:** disinfection, sanitization, onychomycosis, tinea pedis, resistance

## Abstract

Fungal infections of the feet, such as athlete’s foot and toenail fungus, are common and often come back even after treatment. One major reason is that fungus can survive in everyday items like shoes, socks, towels, and even on floors and laundry baskets, leading to reinfection. This review looked at how effective different cleaning and disinfecting methods are at killing these fungi. We reviewed both older and newer strategies, including hot water washing, chemical sprays, ultraviolet light, ozone, and antimicrobial fabrics, to see which ones work best and are safe and practical for everyday use. We found that no single method works for everything. For example, hot water kills fungus on socks but can damage some materials; sprays work on shoes but must be used correctly. Special care is also needed when cleaning tools like nail clippers. The study highlights the importance of combining medical treatment with regular cleaning of personal and shared items to prevent the fungus from spreading or coming back. These findings can help healthcare workers and the public take better steps to stop the cycle of infection and reduce the risk of fungal resistance to medications.

## 1. Introduction

Superficial fungal infections of the feet, including tinea pedis (athlete’s foot) and onychomycosis, represent a significant global public health concern, affecting millions of individuals annually [1,2]. These infections are notably recurrent, with reinfection often traced to contaminated personal items and shared environments [3,4,5]. Items such as shoes, socks, textiles, laundry baskets, and communal surfaces, including floors, mats, and shower stalls, serve as persistent reservoirs for fungal pathogens such as dermatophytes and yeasts [5]. This underscores the urgent need for comprehensive and effective disinfection and sanitization strategies across both personal and communal settings [5]. 

Although clinical management has traditionally focused on pharmacological treatments, the growing emergence of antifungal resistance increasingly challenges this approach [6]. The repeated or improper use of topical and systemic antifungals, especially in chronic or recurrent infections, exerts selective pressure that fosters the development of resistant fungal strains [6]. Environmental reservoirs of infection, when left unaddressed, perpetuate a cycle of reinfection and treatment failure which further exacerbates resistance [5,6]. Alarmingly, awareness of the environmental dimension of transmission remains limited among healthcare professionals [5,7,8]. Routine laundering may be insufficient to eliminate fungal organisms from fabrics, and even visibly clean footwear or laundry containers may harbor viable fungal spores [5,7,8]. As a result, individuals often unknowingly auto-inoculate or transmit resistant strains to others, particularly in high-risk communal environments such as households, locker rooms, public pools, other recreational areas [7,9,10,11], as well as second-hand clothing.

The purpose of this review is to critically assess both historical and emerging methods for disinfecting footwear, socks, foot care instruments, laundry receptacles, and related textiles. It synthesizes current evidence on the efficacy of these interventions, discusses their practical limitations, and identifies gaps in evidence for existing disinfection protocols. By doing so, this review aims to equip healthcare providers, infection control specialists, and researchers with the knowledge needed to implement effective environmental hygiene strategies. Such interventions are essential not only for preventing reinfection and breaking transmission cycles but also for curbing the growing threat of antifungal resistance.

## 2. Materials and Methods

A narrative literature review was conducted using three primary databases: PubMed, Scopus, and Web of Science. The search included peer-reviewed articles published between 1938 and 2025 to capture both historical sanitization practices and emerging technologies.

Search terms were selected to encompass the intersection of sanitization methods and superficial fungal infections, particularly those affecting the feet. Keywords included the following:Footwear sanitization, sock disinfectants;Tinea pedis, onychomycosis, superficial mycoses;UV, ozone, laundering, antifungal materials.

The following is a sample search string for Web of Science: TS=(shoe OR sock OR footwear OR textile* OR instrument*) AND TS=(sanitize* OR disinfect* OR steriliz* OR decontaminat*) AND TS=("tinea pedis" OR "athlete’s foot" OR "superficial mycoses" OR "fungal infection*" OR dermatophyte*) AND TS=(foot OR feet OR plantar OR podiatric)

Inclusion criteria included the following:Investigated or reported on sanitization/disinfection of footwear, socks, textiles, or podiatric instruments;Focused on fungal pathogens relevant to superficial foot infections;Reported on efficacy outcomes or fungal reduction.


Exclusion criteria included the following:
Addressed only pharmacologic treatment without mention of environmental sanitization;Focused on systemic or non-dermatological fungal infections;Were not available in English.

Titles and abstracts were screened for relevance. Full texts of eligible studies were reviewed to extract data on sanitization methods, target organisms, outcomes, and methodological quality. Studies were categorized into thematic sections: physical methods, chemical disinfectants, UV and ozone technologies, laundering practices, antimicrobial textiles, and sterilization of instruments.

## 3. Results

This review evaluated a range of disinfection and sanitization strategies for preventing recurrent foot fungal infections. The methods were categorized into chemical disinfectants, basic approaches, ultraviolet irradiation, ozone- and gas-based methods, laundering, antimicrobial materials, and sterilization of foot care instruments. Each method varied in its mechanism of action, efficacy, limitations, and practicality in real-world applications. Data reported in this section on patient-accessible disinfection methods is summarized in Table 1.

### 3.1. Chemical Disinfectants

Several chemical agents have demonstrated antifungal activity on textiles, though each comes with trade-offs related to safety, efficacy, or practicality.

Formaldehyde, one of the earliest agents used for shoe disinfection, is effective across a wide fungal spectrum through its protein cross-linking mechanism [12]. Despite its efficacy, its use has declined significantly due to toxicity, strong odor, and potential carcinogenicity [12]. 

Chlorine-based agents, particularly sodium hypochlorite (household bleach), remain a cornerstone of fungal decontamination [13]. Sodium hypochlorite exerts fungicidal effects via oxidative damage to cell components [14]. In textile decontamination models, 1:10 diluted bleach (5% to ~0.5%) consistently achieved 100% sporicidal activity against *Trichophyton (T.) mentagrophytes* with 10-min contact times [13]. Furthermore, in vitro studies demonstrate that a 1:100 dilution of Clorox bleach achieved complete inhibition of multiple *Candida* species within 30–60 s, including *C. albicans, C. tropicalis,* and *C. glabrata* [15]. However, bleach has several limitations: it must be freshly diluted to maintain potency, it is corrosive to metals and fabrics, and may pose respiratory risks if not used in well-ventilated areas [13,16]. 

Hypochlorous acid (HOCl), a weaker acid derived from chlorine, offers a less corrosive alternative to bleach. [17] Studies suggest that stabilized HOCl formulations (200 ppm solutions) have high fungicidal efficacy against dermatophytes and *C. albicans* with reduced toxicity and greater surface compatibility compared to traditional bleach [17]. 

**Table 1 biology-14-01016-t001:** Summary of patient-accessible disinfection and sanitization methods for preventing recurrent fungal foot infections, including mechanisms, efficacy, and limitations.

Category	Method	Study Type	Species Investigated	Mechanism of Action	Efficacy Summary	Limitations	References
Chemical Disinfectants	Formaldehyde	In vitro	*T. interdigitale*	Protein cross-linking and denaturation	Historically used for disinfection.	Toxicity, strong odor, carcinogenic potential	Berberian, 1938 [12]
	Chlorine-based (e.g., bleach)	In vitro	*T. mentagrophytes* *C. albicans*	Oxidative damage	Widely recommended for disinfecting hard surfaces.	Corrosive; may damage materials.	Moriello et al., 2013 [13], Estrela et al., 2002 [14]
	Terbinafine Spray	In vitro	*T. mentagrophytes* *T. rubrum* *T. tonsurans*	Inhibits ergosterol synthesis in fungal membranes	1% spray effective against *T. rubrum* on shoe insoles.	Needs repeated application; potential for resistance	Gupta et al., 2001 [18]
	Quaternary Ammonium Soaks	In vitro	*T. rubrum,* *T. interdigitale,* *T. indotineae*	Disrupts membranes, denatures proteins	Fungicidal with adequate contact time; used in healthcare settings.	May cause skin irritation; risk of improper dilution	Skaastrup et al., 2022 [8]
Basic Methods	Boiling	In vivo	*T. mentagrophytes*	Thermal denaturation of fungal proteins	Effectively removes dermatophytes from certain footwear.	Not suitable for all materials; may damage footwear	Tanaka et al., 2006 [19]
	Sun Exposure	In vivo	*T. rubrum* *T. mentagrophytes* *Scopulariopsis brevicaulis* *Aspergillus* spp. *Hendersonula toruloidea*	UV radiation and heat reduce fungal viability	3-day sun exposure reduces contamination in socks.	Dependent on sun intensity, duration, and material	Amichai et al., 2014 [20]
UV Irradiation	UVC Devices	In vitro	*T. rubrum* *T. mentagrophytes*	DNA damage via UVC-induced thymine dimer formation	UVC reduced fungal burden by up to 85% in contaminated shoes; ≥ 0.5 J/cm^2^ at 280 nm LED fully inhibited *T. rubrum.*	Limited penetration; cannot pass through nail	Cronin et al., 2014 [21], Ghannoum et al., 2012 [22]
Gas-Based Sanitization	Ozone	In vitro	*T. rubrum* *T. mentagrophytes* *M. canis* *A. flavus* *A. niger* *Penicillium* *Alternaria* *Candida* *Rhodotorula*	Oxidative stress leading to cell damage	Ozone gas fully eliminated *T. rubrum* after 120 s exposure; effective against dermatophytes and filamentous molds.	Less effective against yeasts; efficacy depends on exposure time	Djaroud et al., 2023 [23]
Laundering	Hot Water (≥60 °C, 45 min)	In vitro	*T. rubrum,* *C. albicans*	Thermal inactivation	Eliminates *T. rubrum* and *C. albicans* from clothing.	Not fabric-safe; high energy consumption	Hammer et al., 2011 [24]
	Warm Water (≥30 °C, 10 min)	In vitro	*T. rubrum,* *C. albicans*	Thermal inactivation	Removes *C. albicans*, but not effective for *T. rubrum* spores.	Spores remain viable	Hammer et al., 2011 [24]
Antimicrobial Materials	Silver-Infused Textiles	In vitro	*T. rubrum* *T. mentagrophytes* *C. albicans*	Disrupts respiration and enzymatic processes	Effective against *C. albicans* and dermatophytes; reduced viability on contact.	Skin reactions in sensitive individuals; cost	Hammer et al., 2012 [25]
	Copper-Infused Textiles	In vitro	*T. rubrum* *T. mentagrophytes* *C. albicans*	Disrupts respiration and enzymatic processes	Minimal activity against *C. albicans;* limited benefit observed.	Sensitivity risk; lower efficacy	Hammer et al., 2012 [25]
	Silver-Ion Laundry Machines	In vitro	*T. rubrum* *C. albicans* *A. flavus*	Ions disrupt fungal membranes during rinse cycle	Reduced fungal counts post-wash; ~4 log_10_ CFU/mL reduction for most species tested.	Limited replication; no analysis of cross-contamination; high cost	Jung et al., 2007 [26]

A.: *Aspergillus*, C.: *Candida*, CFU: colony-forming units, M.: *Microsporum*, T.: *Trichophyton*, UV: ultraviolet.

Isopropyl alcohol (IPA) at 70% concentration has shown rapid and broad-spectrum antimicrobial activity, including fungicidal effects against *C. albicans* and *Aspergillus brasiliensis*, even under high organic load conditions [27]. In a comparative in vitro study, 70% IPA demonstrated ≥ 4-log reductions in fungal viability within 1–5 min of exposure across clean and dirty conditions [27]. Its mechanism of action involves rapid protein denaturation and disruption of cell membranes [27]. However, IPA is not sporicidal and has diminished activity below 50% concentration [27]. Regular use of ethanol sprays by climbing clubs and bowling alleys has been shown to effectively inhibit growth of dermatophytes on rental shoes [11]. 

Hydrogen peroxide is a broad-spectrum oxidizing disinfectant that acts by generating reactive oxygen species, damaging fungal cell structures [28]. In a 2013 study from Moriello et al., a 0.5% hydrogen peroxide product (Accel^®^ TB) achieved 100% sporocidal efficacy against *T. mentagrophytes* on contaminated textiles when applied at 5 mL or five sprays with a 10-min contact time [13]. Lower volumes (1 mL or one spray) were less effective, underscoring the need for sufficient surface coverage [13]. 

Terbinafine spray, an over-the-counter antifungal, inhibits ergosterol synthesis in fungal membranes [18]. It has been shown to reduce *T. rubrum* colonization on insoles when applied consistently [18]. Despite its targeted action, challenges include cost, the need for regular application, and limited efficacy on deeply embedded spores [18]. Insufficient dosing also raises the possibility of selecting drug-resistant strains.

Quaternary ammonium (QAC) detergent soaks provide broad-spectrum fungicidal effects by disrupting cell membranes and denaturing proteins [8]. Common in clinical disinfection protocols, these soaks require accurate dilution to prevent skin irritation or buildup on surfaces [8]. Recent disinfection trials have shown that dermatophyte-inoculated socks soaked in a 0.3% dilution of QAC detergent containing alkyl (C12-16) dimethylbenzylammonium chloride (≤1%), didecyldimethylammonium chloride (≤1%), and C12-C14-alkyl(ethylbenzyl)dimethylammonium chloride (≤1%) for 24 h produced a 100% disinfection rate, while soaking in the same solution for 0.5 h and 2 h produced disinfection rates of 46.2% and 84.6%, respectively [8]. The QAC detergent soaks were equally as effective in terbinafine resistant isolates and wild type dermatophytes [8].

### 3.2. Basic Disinfection Approaches

Boiling footwear using water achieves fungal eradication through thermal denaturation of proteins [19]. Studies confirm that boiling is effective against dermatophytes, particularly when used on heat-tolerant items like rubber sandals or athletic sneakers [19]. However, this method is unsuitable for most synthetic or leather-based shoes due to material degradation risks [19]. 

Sun exposure, while convenient and environmentally sustainable, relies on ultraviolet radiation and ambient heat to reduce fungal viability [20]. Its effectiveness varies greatly depending on environmental factors such as sunlight intensity, duration of exposure, and local climate conditions [20]. While some reduction in fungal load is observed on textiles, this method alone may be insufficient for complete disinfection [20]. 

### 3.3. Ultraviolet Irradiation

Emerging physical disinfection technologies show promising efficacy but come with logistical and financial limitations. Ultraviolet-C (UVC) irradiation disrupts fungal DNA and has demonstrated efficacy in reducing surface contamination. UVC-based devices can be used for disinfection of shoes, insoles, and clinical instruments [21,22]. However, its limited penetration depth and potential harm to human skin or eyes require strict safety protocols during use [21,22]. 

### 3.4. Gas-Based Sanitization

Ozone generators kill fungi through oxidative stress, effectively reducing viable spores in contaminated footwear [29]. Through oxidative degradation of cell structures, ozone treatment is known to inhibit fungal growth at various stages [30]. These systems serve as effective adjunctive methods in high-risk populations or shared environments [29]. Nonetheless, concerns include high equipment cost, risk of ozone exposure to users, and the need for controlled ventilation [29]. Recent research has reinforced ozone gas’s potent antifungal activity, especially against filamentous molds and dermatophytes [23]. A study evaluating ten clinical and environmental fungal isolates, including *T. rubrum* and *T. mentagrophytes* found that ozone exposure led to complete inhibition of fungal growth in sensitive strains, with efficacy increasing proportionally to exposure time [23]. Specifically, *T. rubrum* was fully eradicated after 120 s of ozone exposure [23]. 

Nitrogen dioxide (NO_2_) sanitization is a novel approach that disrupts fungal respiration and protein synthesis, likely due to oxidative and hypoxic stress [31]. While early studies have demonstrated potential antifungal activity, this method remains experimental [31]. Widespread adoption is hindered by the need for pressurized systems and a lack of robust clinical data [31]. 

### 3.5. Laundering Protocols

Laundering remains one of the most common sanitation practices, particularly for socks and textiles in direct contact with infected feet. Hot water washing at ≥ 60 °C for at a least 45-min main washing cycle has proven effective in eliminating dermatophytes and *Candida* species from contaminated fabrics [24]. This approach is particularly recommended for socks, bed linens, and towels shared among individuals with tinea pedis or onychomycosis [24]. However, it is energy-intensive and may damage delicate materials [24]. 

Warm water washing at 30 °C for a 10-min main washing cycle, while less energy-intensive, fails to inactivate dermatophyte conidia, though it effectively eliminates *C. albicans* [24]. As such, it is not recommended as a standalone method for fungal decontamination [24]. 

### 3.6. Antimicrobial Materials

The incorporation of antimicrobial agents into wearable textiles or laundry technology offers a promising, passive means of fungal control. Silver- and copper-infused textiles demonstrate antifungal effects via disruption of fungal respiration and induction of oxidative stress [32]. Silver-infused textiles have shown inhibitory action to both dermatophytes and *C. albicans*; however, copper has only shown low efficacy against *C. albicans* growth [25]. Despite their potential, risks such as skin sensitivity, high cost, and limited product availability restrict their widespread use [25]. 

Silver-ion laundry machines, which release antimicrobial silver ions during washing and rinse cycles, offer an automated and user-friendly approach to reducing fungal loads on clothing [26]. A study by Jung et al. (2007) demonstrated a reduction of over 4 log_10_ CFU/mL in several fungal species, including *T. rubrum* and *C. albicans*, when using the silver-ion machine with detergent [26]. However, methodological limitations in quantifying filamentous fungi, particularly the reliance on colony-forming units (CFU/mL), which may not accurately capture hyphal fragments or non-culturable forms, raise questions about the robustness of the findings [26]. Additionally, the study did not assess whether fungi were transferred to other ‘clean’ fabrics within the same laundry load, leaving the potential for cross-contamination unaddressed [26]. While initial results are promising, the lack of standardized fungal counting protocols and incomplete contamination control limit confidence in broad implementation.

### 3.7. Sterilization of Foot Care Instruments

Foot care instruments such as nail clippers, callus removers, and curettes are frequent vectors for cross-contamination in both home and clinical settings [33,34,35]. Inadequate sterilization poses a major reinfection risk, especially among individuals receiving routine foot care [33,34,35]. 

The Centers for Disease Control and Prevention (CDC) designates steam sterilization as the most reliable method for reusable metal instruments, with effectiveness dependent on precise temperature, pressure, and exposure time [35]. The standard gravity displacement autoclave cycle requires 121 °C (250 °F) at 15 psi for 30 min to ensure destruction of all microbial life [35]. For faster processing in clinical settings, pre-vacuum autoclaves achieve sterilization at higher temperatures of 132–135 °C (270–275 °F) at 30 psi for just 4 min, while flash sterilization, which is reserved for emergencies, uses 132 °C (270 °F) at 30 psi for 3 min [35]. The CDC emphasizes that 121 °C for 30 min remains the gold standard for most non-porous foot care instruments, such as nail clippers and curettes, with studies confirming that properly autoclaved tools show no viable microbial growth post-processing when protocols are followed [35]. 

High-level disinfection (HLD) or sterilization can be achieved with ≥2.4% glutaraldehyde or 1.0% orthophthalaldehyde (OPA), but these require immersion times of 20–90 min for sterilization and adequate ventilation due to toxic fumes [35]. The CDC emphasizes that chemical sterilants are less practical for non-clinical settings due to their occupational hazards and strict handling requirements [35]. 

For heat-sensitive metal instruments, dry heat at 160–170 °C (320–338 °F) for 2 h is an effective alternative, though the CDC notes its inefficiency (longer processing times) compared to steam sterilization [35]. 

The CDC does not recommend UV light cabinets for sterilization, as they provide only surface-level disinfection and fail to inactivate pathogens in crevices or organic debris [35]. Such methods are unsuitable for reusable tools without prior cleaning and autoclaving [35]. Proper sterilization of instruments is crucial in both preventing initial infection and reducing the risk of reinfection, especially in communal or high-risk care environments.

## 4. Discussion

This review highlights the pivotal role of environmental sanitization in managing and preventing recurrent superficial fungal infections such as tinea pedis and onychomycosis. While pharmacologic treatments remain essential to clinical care, their effectiveness is significantly diminished in the absence of concurrent and consistent disinfection of personal items, textiles, surfaces, and foot care instruments. Reinfection through contaminated environments not only undermines treatment success but also contributes directly to the development of antifungal resistance by sustaining exposure to sub-therapeutic fungal burdens. Suggested textile and surface sanitization techniques for patients are outlined in Figure 1 and Table 2.

In exogenous environments, the presence of dermatophyte arthroconidia—significant infectious propagules formed by the fragmentation of hyphae that may remain stable for up to 5 years—underlies the risk of fomite transmissions especially for *Trichophyton* species [36]. Other types of asexual spores produced by dermatophytes include macroconidia, which are large and septate conidia resistant to digestion by arthropods, as well as microconidia, which are smaller and unicellular conidia [36]. Upon adherence to keratinized tissues (skin, hair, nails), arthroconidia undergoes germination forming fungal hyphae, which then invades the non-viable stratum corneum [36]. As infection progresses, more arthroconidia are produced enabling the spread of infection to other anatomical locations, or onto surfaces and fomites, thus continuing the cycle [36].

Among disinfectants experimentally tested against arthroconidia suspensions free from hyphal elements, *T. mentagrophytes* arthroconidia appeared more resistant than those of *T. rubrum* and *T. tonsurans* against phenol, sodium dodecyl sulfate and quaternary ammonium compounds [18]. Consequently, a longer exposure time may be required to achieve fungicidal effects, while chlorine and terbinafine were more effective; the use of sub-therapeutic doses of terbinafine, however, carries a risk of resistance selection amidst the current “epidemic-like” scenario concerning *T. indotineae* [18,37]. In another study, the sporicidal activity of quaternary ammonium compounds against *T. mentagrophytes* and *Microsporum canis* was shown to be dependent on volume or the number of sprays [13]. When exposure time was limited to 1 or 5 min, hydrogen peroxide was shown to be ineffective against non-dermatophyte *Aspergillus* spores [27]. Regarding non-chemical methods, ultraviolet irradiation was shown to be effective against *T. rubrum* and *T. mentagrophytes* conidia suspensions; however, this method requires optimized wavelength (280 nm), power density (1.8 mW/cm^2^), and fluence (3.1 J/cm^2^) settings for efficacy [21,22]. For ozone gas, a single exposure was only partially effective against *T. mentagrophytes* conidia suspensions, whereas 100% inhibition was observed for *T. rubrum* [29]. Taken together, these in vitro observations suggest species-dependent conidia susceptibility to various disinfectants. These observations, however, are limited due to the use of a small number of laboratory isolates. Further studies are warranted to validate and optimize the real-world applications of disinfectants, especially given the emergence of new dermatophyte species or strains from the *T. mentagrophytes* complex with decreased antifungal susceptibility.

The findings from this review indicate that no single sanitization method is universally effective across all materials and use-cases. Chemical disinfectants, such as sodium hypochlorite, hydrogen peroxide, isopropyl alcohol, and quaternary ammonium compounds, offer broad antifungal activity but require careful consideration of concentration, contact time, and material compatibility. Although chlorine bleach remains a gold-standard disinfectant, its corrosive nature and respiratory risks limit its use on sensitive items. 

Physical methods, such as boiling and high-heat laundering (≥60 °C for ≥45 min), are highly effective against dermatophytes and yeasts on socks and washable textiles. However, these methods are unsuitable for delicate fabrics or most types of footwear. Sunlight exposure, while accessible and environmentally friendly, offers inconsistent efficacy due to variable ambient conditions.

Ultraviolet-C (UVC) and ozone-based technologies show strong in vitro antifungal effects and hold promise for disinfecting shoes and surfaces. However, their cost, safety considerations, and lack of standardization limit widespread adoption. Similarly, silver- and copper-infused textiles and silver-ion laundry machines demonstrate antifungal potential, though issues of skin sensitivity, cost, and inconsistent study designs temper enthusiasm for their routine use.

Sterilization of foot care instruments, including nail clippers and curettes, is a critical but often overlooked practice. Autoclaving remains the gold standard due to its ability to eliminate all microbial life when properly applied. Chemical sterilants and dry heat offer alternatives for non-steam-compatible tools but are less practical for routine home use. Importantly, the continued use of UV cabinets without cleaning or autoclaving is insufficient and potentially misleading, as they fail to sterilize creviced or contaminated instruments.

To reduce the risk of reinfection, patients should adopt a multi-pronged hygiene strategy that includes the following:Washing socks, towels, and bed linens in hot water cycles (≥60 °C for 45+ min);Using chemical disinfectants or sprays on shoes and insoles, ensuring proper contact time and ventilation;Periodically exposing items to direct sunlight where feasible;Cleaning and disinfecting shared or high-contact surfaces, including shower floors, laundry baskets, yoga mats, and locker room benches, using bleach, QACs, or hydrogen peroxide-based cleaners;Avoiding shared foot care instruments and sterilizing personal tools regularly;Replacing or rotating contaminated socks and shoes during treatment;Considering antimicrobial textiles or silver-ion laundry appliances where appropriate.

Failure to address these environmental reservoirs not only leads to high reinfection rates but also places evolutionary pressure on fungal populations, facilitating the emergence and persistence of antifungal-resistant strains. This is particularly concerning for immunocompromised individuals or those undergoing long-term antifungal therapy, where treatment options may become increasingly limited.

## 5. Conclusions

Ultimately, the available literature supports the need for a comprehensive, evidence-based sanitization framework that combines patient education, routine hygiene practices, and the integration of validated disinfection technologies. This review is limited by the lack of clinical validation studies examining differences in real-world disinfection approaches in terms of relapse, recurrence and familial transmissions. Disinfection/decontamination methods remain an underexplored research area for dermatophytosis management and deserves increased attention in view of the newly emerged species or clonal offshoots of the *T. mentagrophytes* complex with epidemic potential and antifungal resistance propensity (e.g., *T. indotineae*, *T. mentagrophytes* ITS genotype VII). Clinical and public health messaging should emphasize the environmental component of fungal transmission with the same urgency as pharmacologic adherence. 

## Figures and Tables

**Figure 1 biology-14-01016-f001:**
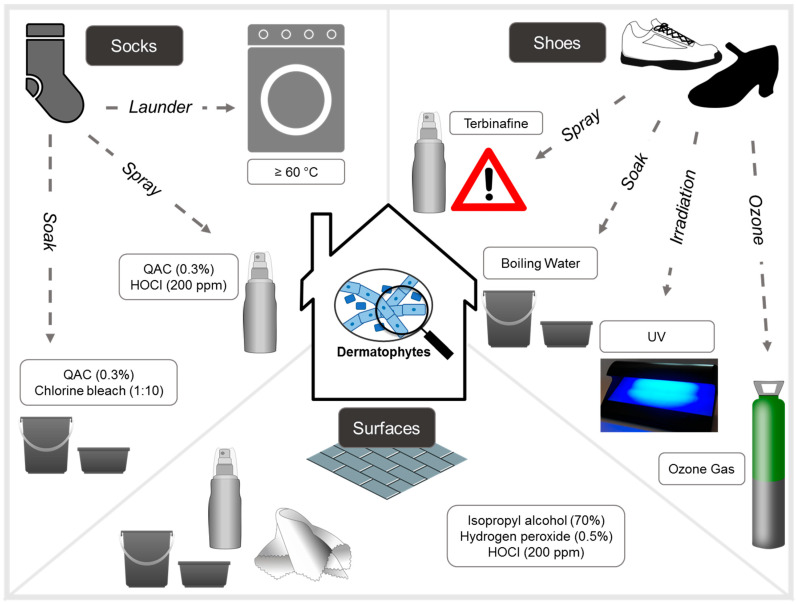
Summary of experimental methods tried for the disinfection of household items, surfaces, shoes and socks for dermatophytic fungi. Note that authors caution against the use of terbinafine spray as sub-therapeutic doses has been linked to antifungal drug resistance selection.

**Table 2 biology-14-01016-t002:** Recommended disinfection methods for common fomites.

Item	Disinfection Method	Agent/Tool	Exposure Conditions	Efficacy	Limitations/Considerations
White Cotton Socks	-Laundering (thermal) -Chemical soak -Surface spray	-Hot water (≥60 °C) [24] -Chlorine bleach (1:10) [13] -QAC (0.3%) [8] -Hydrogen peroxide (0.5%) [13]	-60 °C main wash cycle ≥ 45 min [24] -Bleach 10 min soak [13] -QAC 24 h soak [8] -Peroxide: 5 sprays, 10 min [13]	-100% kill of dermatophytes and *Candida* with hot wash and bleach [13,24] -QAC: 100% kill after 24 h [8] -Peroxide: 100% sporicidal with 5 sprays [13]	-Bleach degrades fabric, strong odor [13] -QAC requires long soaking [8] -Peroxide needs full coverage [13]
Colored Socks	-Laundering (thermal) -Chemical soak/spray	-Hot water (≥60 °C) [24] -QAC (0.3%) [8] -Hypochlorous acid (HOCl 200 ppm) [17]	-Same as above	-Same QAC efficacy as white socks [8] -HOCl: High fungicidal action against *C. albicans* and dermatophytes [17]	-Avoid chlorine bleach to preserve fabric color [17] -HOCl is safer, less corrosive, but may be less accessible [17]
Wool Socks	-Chemical soak/spray only (no hot water or bleach)	-QAC (0.3%) [8] -HOCl (200 ppm) [17]	-QAC 24 h soak [8] -HOCl 10–15 min spray or soak [17]	-QAC: 100% kill after 24 hrs [8] -HOCl: Effective with minimal textile damage [17]	-Avoid hot water or bleach to prevent wool damage [24] -Must ensure full immersion or spray coverage [17]
Shoes	-Antifungal spray -UV-C light -Ozone gas -Boiling (for some)	-Terbinafine spray [18] -UV-C exposure [21,22] -Ozone generator [23,29] -Boiling water [19]	-Daily application for spray [18] -UV-C for 5–15 min [21,22] -Ozone: ≥120 s [23] -Boiling: 100 °C, 5–10 min [19]	-Terbinafine reduces *T. rubrum* load [18] -UV-C and ozone: 100% inhibition in studies [21,22,23] -Boiling kills spores [19]	-UV-C requires safety precautions and surface contact [21,22] -Ozone needs ventilation [29] -Boiling damages non-rubber shoes [19]
Surfaces (Floors, mats, laundry receptacles)	-Alcohol-based disinfection -Hydrogen peroxide -HOCl spray	-Isopropyl alcohol 70% [27] -Hydrogen peroxide 0.5% [13] -HOCl 200 ppm [17]	-IPA: ≥1–5 min contact [27] -Peroxide: 5 sprays, 10 min contact [13] -HOCl: 10 min soak/spray [17]	-IPA: ≥4-log reduction in fungal load [27] -Peroxide: 100% sporicidal with 5 sprays [13] -HOCl: High efficacy [17]	-IPA not sporicidal, evaporates quickly [27] -Peroxide needs proper dosage [13] -HOCl is safer but less available [17]

## Data Availability

No new data created.

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
