# Peer review of "Hygiene Practices Against Dermatophytic Fungi: A Review of Strategies to Combat Antifungal Resistance"

_biology, 2025, doi:10.3390/biology14081016_

Round 1
Reviewer 1 Report
Comments and Suggestions for Authors
-
Missing a mechanistic model
There is no clear framework that connects spore resistance mechanisms (e.g., metabolic quiescence, hydrophobicity, biofilm protection) to the efficacy of each disinfectant. A conceptual figure could improve this. -
Inadequate clinical correlation
Although the review emphasizes clinical implications, there is no systematic analysis of clinical outcomes in relation to sanitization strategies. For example, are there RCTs or cohort data showing reduced relapse with environmental disinfection? The lack of patient-centered outcomes limits utility. -
Too much emphasis on in vitro data
Multiple methods report excellent fungicidal outcomes in vitro, yet the translation to clinical practice is nontrivial, especially with items like shoes or textured materials. The authors should caution more explicitly about this gap. -
Overlooked emerging resistance mechanisms
The paper could benefit from a deeper discussion on terbinafine resistance pathways (e.g., squalene epoxidase mutations) and how environmental exposure (sublethal concentrations on surfaces) could be contributing. -
No discussion of environmental fungal ecology
A major omission is the lack of reference to environmental survival dynamics—e.g., spore adherence to surfaces, fomite colonization cycles, or how humidity and temperature affect fungal viability. -
Practical feasibility not evaluated adequately
Many methods—like 24-hour QAC soaks or silver-ion laundry machines—are impractical for real-world users. A cost/accessibility matrix or feasibility rating would add clinical relevance.
Author Response
Reviewer 1:
- Missing a mechanistic model
There is no clear framework that connects spore resistance mechanisms (e.g., metabolic quiescence, hydrophobicity, biofilm protection) to the efficacy of each disinfectant. A conceptual figure could improve this.
Thank you for sharing your insights. While we believe that resistance of infectious propagules (e.g., arthroconidia) in the environment is significant when considering disinfection approaches, we could not find published studies to expand on this point. To our knowledge, none of the disinfectants included in this review were evaluated against resistance mechanisms in dermatophyte spores (macroconidia, microconidia, arthroconidia). Hence, a broader framework is utilized here: reducing the overall dermatophyte viability with the use of disinfectants.
- Inadequate clinical correlation
Although the review emphasizes clinical implications, there is no systematic analysis of clinical outcomes in relation to sanitization strategies. For example, are there RCTs or cohort data showing reduced relapse with environmental disinfection? The lack of patient-centered outcomes limits utility.
We agree that clinical data should be the cornerstone when recommending disinfection approaches. To our knowledge, none of the studies so far have evaluated disinfection procedures against frequencies of relapses or recurrences in dermatophytosis patients. All of the work included in this review were done in vitro (see Table 1). We have expanded the Discussion section to highlight the roles of arthroconidia in the dermatophyte lifecycle, as well as available evidence on the efficacy of disinfectants against arthroconidia (lines 278-309).
- Too much emphasis on in vitro data
Multiple methods report excellent fungicidal outcomes in vitro, yet the translation to clinical practice is nontrivial, especially with items like shoes or textured materials. The authors should caution more explicitly about this gap.
We agree that experiments often lack clinical translation. Research on disinfectants is often overlooked, but has the potential to significantly improve the management of dermatophytosis owing to the current high risks of relapse and recurrence. We have highlighted this gap in the Conclusions section (lines 363-369).
- Overlooked emerging resistance mechanisms
The paper could benefit from a deeper discussion on terbinafine resistance pathways (e.g., squalene epoxidase mutations) and how environmental exposure (sublethal concentrations on surfaces) could be contributing.
We agree that sub-therapeutic doses of terbinafine could contribute to the global rise in terbinafine resistance. We have cautioned against the use of terbinafine as a disinfectant (lines 159-160; 292-294; Figure 1).
- No discussion of environmental fungal ecology
A major omission is the lack of reference to environmental survival dynamics—e.g., spore adherence to surfaces, fomite colonization cycles, or how humidity and temperature affect fungal viability.
We have expanded the Discussion section to introduce the survival dynamics of dermatophytes in exogenous environments (Lines 278-287).
- Practical feasibility not evaluated adequately
Many methods—like 24-hour QAC soaks or silver-ion laundry machines—are impractical for real-world users. A cost/accessibility matrix or feasibility rating would add clinical relevance.
We fully agree that there are practical hurdles to implementing these newer methods. However, analysis on cost and accessibility would differ significantly by country as well as among historically underserved populations; hence, it is beyond our ability to incorporate such analysis in this review. Furthermore, as previously noted, there is a common lack of clinical validation for these disinfection methods. In view of the current landscape of antifungal resistance in dermatophytes, with an increasing health and socioeconomic burden, it is our hope that the urgency will incentivize public health actions that will make effective disinfection methods more readily accessible.
Reviewer 2 Report
Comments and Suggestions for Authors
Dear editor and authors,
I consider that the paper is well structured and well written. The paper is a very important contribution to the scientific literature, so I suggest its acceptance for publication. However, I suggest reviewing the following comments to improve some areas.
1. It is recommended to include the type of study in the title.
2. In table 1: the name of the author and the date of publication are usually added in the references section.
3. I recommend adding a section on the strengths and limitations of the paper in the discussion.
Author Response
- It is recommended to include the type of study in the title.
Thank you for your suggestion, the title has been revised.
- In table 1: the name of the author and the date of publication are usually added in the references section.
Thank you for your suggestion, the table has been revised accordingly.
- I recommend adding a section on the strengths and limitations of the paper in the discussion.
Thank you for your suggestion. We have revised the Discussion and Conclusion sections regarding the limitations of this review (Lines 306-309; Lines 363-369).
Reviewer 3 Report
Comments and Suggestions for Authors
1 Methodology of the review
I appreciate that you based your search on three reputable databases—PubMed, Scopus, and Web of Science—and that you limited the publication window to 2020-2025. The manuscript lists sample keywords and provides the complete Web of Science search string, facilitating replication. I also appreciate the general inclusion and exclusion criteria you outlined, clarifying that only studies on shoe, textile, or instrument disinfection, as well as those on superficial infections, were considered, while purely pharmacological papers and non-English publications were excluded.
However, no information is provided on the total number of records identified, excluded, and finally included, nor is a PRISMA diagram included. A detailed risk-of-bias assessment of the source studies is also missing, as is a clear statement on whether the review is narrative or systematic. For complete transparency, I recommend adding a PRISMA flowchart, a study quality table, and a declaration of protocol status (ideally with a PROSPERO registration number, if you decide to register).
2 Self-citations
Analysis of the reference list shows that nine of the thirty-five entries—approximately 26%—are papers with Dr. Gupta as the first author. The self-citations are thematically justified, covering onychomycosis, shoe disinfection, dermatophyte resistance, and the effects of terbinafine and ozone. Nevertheless, the proportion exceeds the commonly accepted 20 % threshold. I therefore suggest reducing the share of self-citations, especially in sections that rely solely on your data (e.g., ozone treatment and resistance), by adding findings from independent research groups. This will strengthen the objectivity of the manuscript.
3 Balance and quality of sources
Most cited studies are from the past five years and appear in reputable journals; guidelines from the CDC and IPAC Canada are also referenced, which is a significant strength. Still, there is a lack of solid clinical or epidemiological evidence demonstrating the real-world impact of hygiene practices on reducing recurrence or transmission. Please expand the bibliography to include randomized trials or extensive multicenter observational studies.
4 Tables and data presentation
Tables 1 and 2 are clear and contain key elements, including the mechanism of action, efficacy, limitations, and references. I propose adding extra columns for: study type (in vitro, in vivo, clinical), number of publications supporting each method, and an approximate evidence level (e.g., a simplified A/B/C or GRADE scale). Mark rows based solely on the authors’ results with an asterisk to help readers judge the independence of the data.
5 Limitations and future directions
Create a dedicated subsection, “Limitations & Future Directions”, that summarizes the main weaknesses of the existing evidence: the dominance of in vitro studies, the lack of protocol standardization (concentration, contact time, substrate/medium), and limited clinical data. Also highlight practical barriers (ozone toxicity, material degradation, and the cost of modern devices) and issue a specific call for randomized interventional trials.
6 Strengthening the resistance discussion
Consider expanding the discussion with the latest epidemiological data on the Trichophyton indotineae strain and its global spread. Explain mechanistically how inadequate environmental hygiene promotes selection of resistant strains, primarily through repeated reinfection cycles and sub-optimal treatment.
7 Editorial corrections
Replace “buying recycled clothes” with the more precise “second-hand clothing”.
8 Practical materials
Finally, attach a simple infographic or checklist for patients and staff, including minimum washing temperatures, recommended chemical concentrations, and required exposure times for UV or ozone treatment.
Comments on the Quality of English LanguageReplace “buying recycled clothes” with the more precise “second-hand clothing”. Also, spell out the abbreviations QAC, HLD, and OPA at their first appearance.
Author Response
1 Methodology of the review
I appreciate that you based your search on three reputable databases—PubMed, Scopus, and Web of Science—and that you limited the publication window to 2020-2025. The manuscript lists sample keywords and provides the complete Web of Science search string, facilitating replication. I also appreciate the general inclusion and exclusion criteria you outlined, clarifying that only studies on shoe, textile, or instrument disinfection, as well as those on superficial infections, were considered, while purely pharmacological papers and non-English publications were excluded.
However, no information is provided on the total number of records identified, excluded, and finally included, nor is a PRISMA diagram included. A detailed risk-of-bias assessment of the source studies is also missing, as is a clear statement on whether the review is narrative or systematic. For complete transparency, I recommend adding a PRISMA flowchart, a study quality table, and a declaration of protocol status (ideally with a PROSPERO registration number, if you decide to register).
Thank you for your comments. In our view, this work does not qualify as a systematic review since most of the studies in this research area are limited to in vitro experiments on a small number of laboratory isolates that precludes risk-of-bias assessments. While we formulated a search strategy and used three databases, we have emphasized that this work constitutes a narrative review (Line 79) since no clinical studies to our knowledge have examined disinfection methods.
2 Self-citations
Analysis of the reference list shows that nine of the thirty-five entries—approximately 26%—are papers with Dr. Gupta as the first author. The self-citations are thematically justified, covering onychomycosis, shoe disinfection, dermatophyte resistance, and the effects of terbinafine and ozone. Nevertheless, the proportion exceeds the commonly accepted 20 % threshold. I therefore suggest reducing the share of self-citations, especially in sections that rely solely on your data (e.g., ozone treatment and resistance), by adding findings from independent research groups. This will strengthen the objectivity of the manuscript.
Thank you for your comments. We recognize the importance of including findings from a diverse selection of research groups. Following our revisions, 6 of 37 cited publications (16.2%) are papers with Dr. Gupta listed as an author.
3 Balance and quality of sources
Most cited studies are from the past five years and appear in reputable journals; guidelines from the CDC and IPAC Canada are also referenced, which is a significant strength. Still, there is a lack of solid clinical or epidemiological evidence demonstrating the real-world impact of hygiene practices on reducing recurrence or transmission. Please expand the bibliography to include randomized trials or extensive multicenter observational studies.
Thank you for your comments. Disinfection remains an underexplored research area for the management of dermatophytosis as most studies are in vitro experiments. To our knowledge, no clinical trials have been conducted to date examining the real-world use of disinfectants on reducing the risk of relapse, recurrence or familiar transmission (Lines 363-369).
4 Tables and data presentation
Tables 1 and 2 are clear and contain key elements, including the mechanism of action, efficacy, limitations, and references. I propose adding extra columns for: study type (in vitro, in vivo, clinical), number of publications supporting each method, and an approximate evidence level (e.g., a simplified A/B/C or GRADE scale). Mark rows based solely on the authors’ results with an asterisk to help readers judge the independence of the data.
A Study Type column has been added to Table 1. As noted above, a key limitation of this review is that the current research progress on dermatophyte disinfection is mainly in vitro experiments with a low level of evidence. Regardless of the differences in efficacy observed across studies, a prospectively designed clinical trial is warranted before any solid clinical recommendations can be made. Although we recognize that some of the works included in this review involved authors’ prior publications, we believe that further enumeration is unnecessary given that all studies included in this review share a common limitation.
5 Limitations and future directions
Create a dedicated subsection, “Limitations & Future Directions”, that summarizes the main weaknesses of the existing evidence: the dominance of in vitro studies, the lack of protocol standardization (concentration, contact time, substrate/medium), and limited clinical data. Also highlight practical barriers (ozone toxicity, material degradation, and the cost of modern devices) and issue a specific call for randomized interventional trials.
Thank you for your comments. We agree that each method should be examined critically in terms of practicality, costs, and variations in experimental design. We have highlight limitations/considerations of each method in Table 2. A need for clinical data is highlighted in the Conclusions section (Lines 363-369).
6 Strengthening the resistance discussion
Consider expanding the discussion with the latest epidemiological data on the Trichophyton indotineae strain and its global spread. Explain mechanistically how inadequate environmental hygiene promotes selection of resistant strains, primarily through repeated reinfection cycles and sub-optimal treatment.
Thank you for your suggestion. We agree that environmental disinfection deserves increase attention given the global emergence of antifungal resistance, and specifically T. indotineae. Inadequate use of terbinafine at sub-therapeutic doses certainly poses a risk for resistance development (Lines 292-294; Figure 1). However, the evolution of T. indotineae remains elusive at present. To our knowledge, epidemiological studies have not been to able to pinpoint the exact cause of the current “epidemic-like scenario” in the Indian subcontinent. While over-the-counter medications with suboptimal doses could be one factor, environmental factors such as climate change may also play a role.
7 Editorial corrections
Replace “buying recycled clothes” with the more precise “second-hand clothing”.
Thank you for your suggestion, this has been changed accordingly (Line 68).
8 Practical materials
Finally, attach a simple infographic or checklist for patients and staff, including minimum washing temperatures, recommended chemical concentrations, and required exposure times for UV or ozone treatment.
We agree that a simple infographic would improve the readability of our manuscript. We have added Figure 1 accordingly; details about each methodology are shown in Table 2.
Quality of English: Replace “buying recycled clothes” with the more precise “second-hand clothing”. Also, spell out the abbreviations QAC, HLD, and OPA at their first appearance.
Thank you highlighting this issue. The wording has been replaced to “second-hand clothing” (Line 68), and the abbreviations listed in this comment have been spelt out in full at first mention within the manuscript (Line 161, Line 254, Line 255).
Reviewer 4 Report
Comments and Suggestions for Authors
Check the attached file.

Author Response
Reviewer 4
“The only thing I find missing is that the paper mentions the specific hygiene of the sporulated forms (macroconidia and microconidia) of the different species, perhaps because no studies have been conducted on this subject. However, it would be interesting to know if, in all the data reviewed and found for the different species analyzed, the authors have been able to gather any information on the hygiene of sporulated forms of dermatophytes that can resist some of the treatments mentioned in the paper. If not, at least establish a point of reflection for future studies on the subject.”
Thank you for your comment. We agree that dermatophyte conidia are an important concern with applying disinfection methods. We have expanded the Discussion on this point (Lines 278-309).
Round 2
Reviewer 3 Report
Comments and Suggestions for Authors
Well done!
Well written manuscript. Thank you for your kind response to my comments.
Nothing more to add from my side.